ecology, evolution, genetics

glacial cycles, drainage reversals, phylogeography, sequenced microsatellites, *Percichthys trucha*, glacial refugia

**Author for correspondence:**
Daniel E. Ruzzante
e-mail: daniel.ruzzante@dal.ca

# Multiple drainage reversal episodes and glacial refugia in a Patagonian fish revealed by sequenced microsatellites

Daniel E. Ruzzante[1], Annie P. Simons[1], Gregory R. McCracken[1], Evelyn Habit[2] and Sandra J. Walde[1]

[1]Department of Biology, Dalhousie University, Halifax, Nova Scotia, Canada
[2]Departamento de Sistemas Acuáticos, Facultad de Ciencias Ambientales y Centro EULA, Universidad de Concepción, Concepción, Chile

DER, 0000-0002-8536-8335; GRM, 0000-0002-1701-1529; EH, 0000-0002-5113-5496

The rise of the southern Andes and the Quaternary glacial cycles influenced the landscape of Patagonia, affecting the phylogeographic and biogeographic patterns of its flora and fauna. Here, we examine the phylogeography of the freshwater fish, *Percichthys trucha,* using 53 sequenced microsatellite DNA markers. Fish ($n = 835$) were collected from 16 river systems (46 locations) spanning the species range on both sides of the Andes. Eleven watersheds drain to the Pacific, five of which are trans-Andean (headwaters east of Andes). The remaining five drainages empty into the Atlantic. Three analytical approaches (neighbour-joining tree, hierarchical AMOVAs, Structure) revealed evidence of historic drainage reversals: fish from four of the five trans-Andean systems (Puelo, Futalaufquen/Yelcho, Baker, Pascua) exhibited greater genetic similarity with Atlantic draining systems than with Pacific systems with headwaters west of Andes. Present-day drainage (Pacific versus Atlantic) explained only 5% of total genetic variance, while ancestral drainage explained nearly 27% of total variance. Thus, the phylogeographic structure of *P. trucha* is consistent with episodes of drainage reversal in multiple systems and suggests a major role for deglaciation in the genetic and indeed the geographical distribution of *P. trucha* in Patagonia. The study emphasizes the significant role of historical processes in the current pattern of genetic diversity and differentiation in a fish from a southern temperate region.

## 1. Introduction

A fundamental question in phylogeography and evolutionary and conservation biology is what factors, climatic, geomorphological or other, have influenced the contemporary distribution of genetic diversity and indeed the geographical distribution of species. The glacial cycles of the Quaternary often figure prominently among these factors [1–3]. Our general understanding of the evolutionary influences of these climate cycles has improved dramatically over the last three decades. Initially most phylogeographic studies focused on Northern Hemisphere taxa [4]. More recently, however, a number of studies have examined these questions on Southern Hemisphere species including species from tropical South America [5–8] and its temperate southern cone, Patagonia. In Patagonia, the focus area of the present study, phylogeographic patterns have been described for terrestrial [9–12], coastal marine [13–15]; freshwater [16–23] and diadromous species [24–27].

The processes usually described as the two most important factors in South America are the uplift of the Andes, which began approximately 23 Ma [28], and the glacial cycles of the Quaternary (2.5 Ma–10 000 BP). Their combined influence has affected phylogeographic patterns for many species, including the number and location of glacial refugia and the rates and directions of late and

post-Pleistocene gene flow [12,24,29,30]. Some species survived the last glacial maximum (LGM) in glacial refugia east of the Andes, on the largely unglaciated Patagonian steppe [10,12,17,22,31]. A few cold-adapted species also survived in small refugia west of the Andes within the area mostly covered by glaciers (i.e. south of 42° S; [22,32–34]). Refugia have also been identified west of the Andes north of 42° S where the ice cover during the LGM did not reach the Pacific Ocean [16,22,34,35].

As the climate warmed following the LGM and glaciers began to melt, there were significant changes to the Patagonian landscape, especially along the backbone of the Andes. As the glaciers retreated, large proglacial lakes formed in several places along the eastern flank of the Andes [35–41]. Known palaeolakes include Lake Elpalafquen (41° S), Lake Cari Lafquen, on the Patagonian steppe also at approximately 41° S [34, p. 506], Lake Chalenko, which covered present-day lakes Buenos Aires-General Carrera and Pueyrredón-Cochrane [40–41], Lake Caldenius, encompassing present-day lakes Azara, Belgrano, Mogote, Nansen, Volcán and Burmeister (approx. 48° S) [42–44], and Lake Fuegian, on the island of Tierra del Fuego [44]. After formation, and for perhaps several hundred years, the palaeolakes drained to the Atlantic; ice dams at their western limits prevented westerly flow. As melting continued, however, the western ice barriers were breached, sometimes catastrophically, and flow reversed, draining towards the Pacific [38]. Water levels subsided, leaving the current high elevation Patagonian lakes as remnants of the ancient palaeolakes.

It has been pointed out previously that phylogeographic patterns for *Percicthys trucha* and *Galaxias platei* based chiefly on mitochondrial DNA (mtDNA) polymorphism [16,22], are broadly consistent with post-glacial drainage reversals, but geomorphological evidence for catastrophic drainage reversals is available in detail only for the Baker river valley [38,40]. Here, we revisit the phylogeographic pattern in *P. trucha*, over its entire geographical range on both sides of the Andes (16 drainages and 46 sampling locations) (figure 1) and using a suite of 53 sequenced nuclear microsatellite DNA markers and individuals ($n = 835$). The use of a high number of sequenced nuclear microsatellite markers provides two advantages: (i) it addresses the general concern common to most phylogeographic studies, that patterns based on mtDNA polymorphism reflect the coalescence time and evolutionary history of a single gene, and not necessarily that of the organism a whole; and (ii) their relatively high mutation rate resulting in high polymorphism makes microsatellite markers particularly useful for phylogeographic studies, that both focus on relatively short time frames (e.g. Late Pleistocene–Holocene to present) and examine patterns at a finer geographical scale than may often be feasible with mtDNA. We examined the potential consequences that the major climatic cycles of the Quaternary and the resulting changes in regional geomorphology had on the distribution of genetic diversity, and indeed on the geographical distribution of the species. We assess whether the phylogeographic patterns described for *P. trucha* using mtDNA haplotypes [16] are confirmed by patterns of divergence in genomic DNA, or conversely whether genomic DNA tells a different story regarding the geomorphological history of the region. Our data provide genomic evidence of drainage reversals in several systems for which no detailed geomorphological evidence exists yet, including the Puelo and the Futalaufquen-Yelcho systems in northwestern

Patagonia and the Pascua system in southern Patagonia. These results suggest phylogeographic data can complement and indeed assist in the design of geomorphological studies and vice versa while confirming the role that past processes have played in shaping current species distributions.

## 2. Material and methods

### (a) Species, study sites and sample collection

*Percichthys trucha*, a freshwater fish species native to Patagonia and neighbouring regions to the north, is widely distributed throughout the region. The species is not harvested nor is it the focus of recreational angling, which in Patagonia focuses on introduced salmonids. The species has not ever been the subject of supplementation with hatchery reared fish in any of the river systems visited for this study. Current phylogeographic and genetic diversity patterns can thus be safely considered to be solely the result of natural, biogeographical and geomorphological processes and drivers.

A total of $n = 835$ *Percichthys trucha* collected from 16 systems throughout the geographical range of the species in Patagonia (33° S–50° S; electronic supplementary material, table S1; figure 1) were genotyped for 53 microsatellite markers (electronic supplementary material, table S2). Eleven systems drain into the Pacific Ocean; of which six have their headwaters in Chile (Maipo, Nilahue, Rapel, Biobío, Bueno, Maullín), but the other five have their headwaters east of the Andean highest peaks (Valdivia, Puelo, Futalaufquen/Yelcho, Baker, Pascua). The remaining five systems (Colorado, Negro, Chubut, Chalia and Santa Cruz) have their headwaters east of the Andes and drain into the Atlantic (electronic supplementary material, table S1). Sampling was conducted between 1998 and 2008. Lakes were sampled with gillnets (mesh sizes 30–140 mm) and rivers were sampled by electrofishing. Tissue samples (blood or fin clips) were placed in ethanol and subsequently stored at −20°C.

### (b) DNA extraction

Caudal fin clips and blood samples, both approximately 2 mm³, were digested in proteinase K (Bio Basic Inc., Markham, Ontario, Canada) at 55°C for 8 h. A glassmilk protocol modified from [45] was used to extract the DNA using a Perkin Elmer Multiprobe II plus liquid handling system (Perkin Elmer, Waltham, MA, USA). DNA quality and quantity were checked against a standard using gel electrophoresis on 2% agarose gel.

### (c) Primer design and testing

Information on primer design and testing is available in the electronic supplementary material.

### (d) Genotyping

Genotyping was conducted using MEGASAT [46]. Microsatellite sequence data were checked and the input file was modified as necessary (e.g. changes to flanking regions) to verify that not only were target microsatellite sequences retained, and non-target sequences discarded, but to ensure we were not accidentally discarding useful sequence data, or potentially entire loci. Histogram plots from MEGASAT were manually verified to ensure correct allele call. Microsatellite loci were removed from further analysis if they did not amplify, were too difficult to score accurately, or were monomorphic. In total, 14 out of 75 loci were removed for these reasons. The remaining 61 were amplified (again using two multiplex reactions, this time consisting of 29 and 32 loci, respectively) and sequenced for the remaining samples. Additional loci filtering steps occurred once data had

Proc. R. Soc. B **287**: 20200468

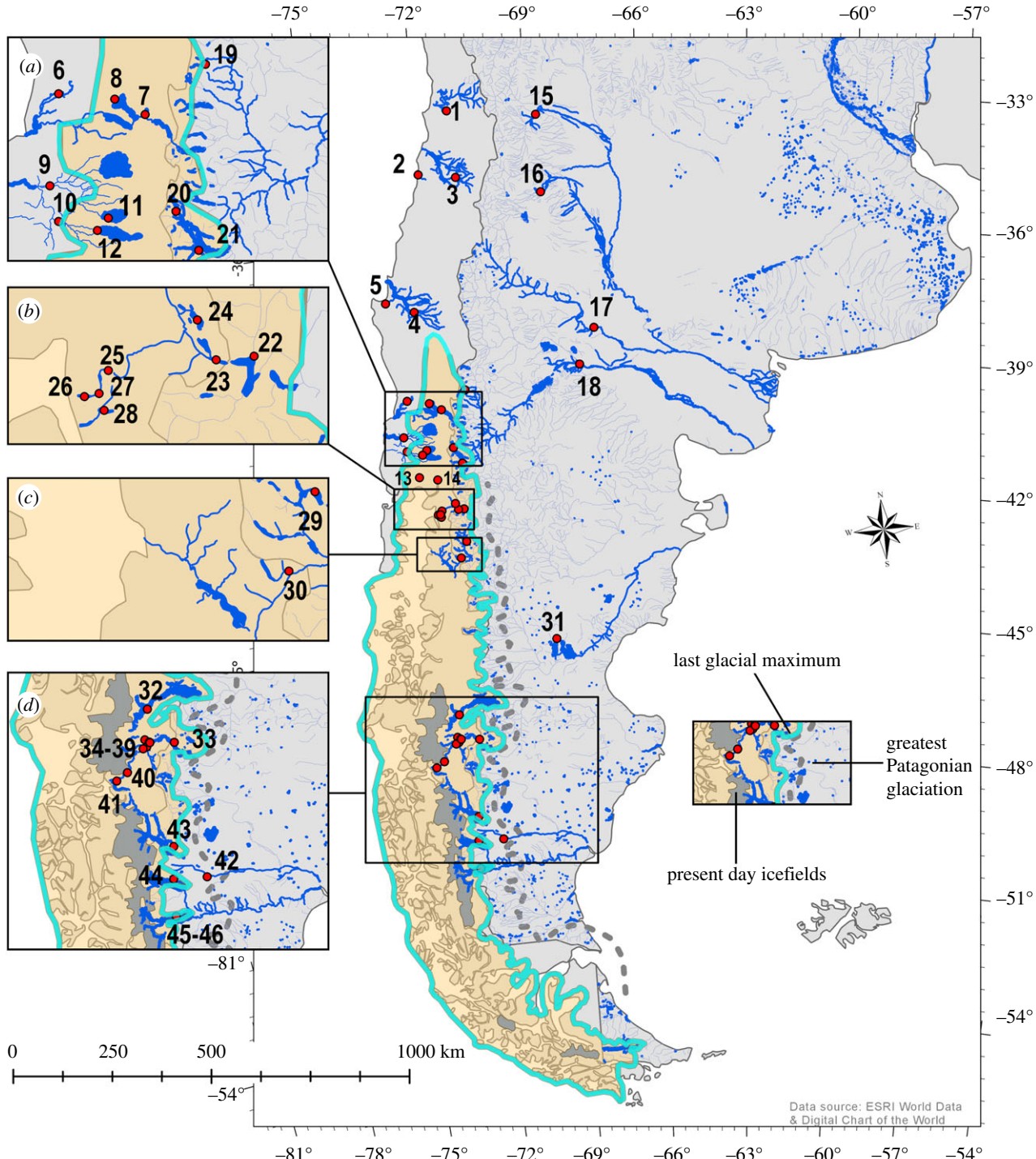

**Figure 1.** *Percichthys trucha* sampling locations. West of the Andes south of 42° S, where the LGM reached the edge of the continental shelf, *P. trucha* is present only in locations associated with trans-Andean systems. Insets: sampling locations in the (*a*) Valdivia, Bueno and Limay/Negro, (*b*) trans-Andean Puelo, (*c*) trans-Andean Futalaufquen/Yelcho, and (*d*) trans-Andean Baker and Pascua systems. (Online version in colour.)

been collected for all 835 individuals. These steps included removing loci which contained null alleles (identified using Microchecker [47]), exhibited greater than 12% missing data, or were sequenced at low depth (less than 50). BayeScan v. 2.1 [48] was run, using default parameters, to identify putative loci under selection. The final number of loci for all subsequent analyses post filtering was 53 (electronic supplementary material, table S1) with an average of 2.05% missing data per locus.

### (e) Phylogeographic reconstruction

We used the program Structure 2.3.4 [49] to assess population structure hierarchically. We first examined the entire dataset. Identified clusters were then independently subject to further

Structure analysis. Structure was run using 500 000 Markov chain Monte Carlo permutations and 100 000 burn-in steps with each *K* value replicated five times. We used the Evanno method [50] as implemented in Structure Harvester v. 0.6.92 [51] to determine the most likely number of clusters. This process was continued up to the identification of individual rivers or lakes. If needed, geographical location was also taken into consideration.

Clumpp 1.1.2 [52] was used to combine the five replicates for the chosen K value into a single output which was then visualized using Distruct 1.1 [53]. Poptree2 [54] was used to create a neighbour-joining phylogenetic tree (with 1000 bootstrap iterations), using Nei's distance (Da), which has a high likelihood of producing a more accurate tree when microsatellite data are used [55]. The phylogenetic tree was then visualized using the

Interactive Tree of Life (iTOL) v. 4 [56]. Pairwise $F_{ST}$ values were not calculated owing to the high variation in sample sizes (electronic supplementary material, table S1), which can bias estimates [57].

Two analyses of molecular variance (AMOVA) were performed using Arlequin 3.5.2.2 [58]. In the first analysis, we grouped collections by contemporary drainage configuration, with one group comprising all systems currently draining into the Pacific Ocean, regardless of headwater location (i.e. whether headwaters are west or east of the Andes), and the second group comprising all systems currently draining into the Atlantic Ocean. In the second AMOVA, we also considered two groups, but this time all systems with headwaters east of the Andes were grouped together, regardless of whether they currently drain into the Atlantic or Pacific Ocean. This grouping is expected to reflect the drainage scenario during and prior the LGM. An increase in the percentage of among-group variance explained under the second scenario would suggest that *P. trucha* populations inhabiting trans-Andean systems (those with headwaters east of the Andes but draining into the Pacific Ocean) are more closely related to other Atlantic draining systems than they are to Pacific draining systems. We subsequently repeated this analysis changing the grouping of one trans-Andean system at a time to examine each system's relative influence on patterns of genetic diversity. Arlequin 3.5.2.2 [58] was used also for testing for linkage disequilibrium (LD).

We acknowledge that some authors [59] consider that all *Percichthys* populations in Chile, including those in the southern Chile, are not *P. trucha,* but a different species entirely, *Percichthys chilensis*, a view the genetic and phylogeographic data we report in this study do not support (see below).

## 3. Results

### (a) Genetic quality control

Although the entire dataset consisted of $n = 835$ individuals genotyped at a panel of 53 microsatellite loci, sample sizes per population were relatively small (mean $n = 18.2$, median $n = 20.5$). We tested for LD between pairs of loci in the nine collections available with $n \geq 30$ (see the electronic supplementary material, table S1). We found no evidence of LD that was significant and consistent across the nine populations tested. We also tested for evidence of selection using all 46 collections, and subsets with relatively large sample sizes. These included the nine collections with $n \geq 30$, the four collections within the Limay-Negro system (all $n \geq 26$) and the two collections within the Maullín system (both $n \geq 27$). While 18, 9, 1 and 0 loci showed up as outliers in each of these tests, respectively, there was no consistency in the identity of the outlier loci across more than two of these four tests. We conclude there is no meaningful evidence of selection in any of these loci and all were retained in all subsequent analyses.

### (b) Neighbour-joining tree

The neighbour-joining tree clusters all Chilean populations north of latitude 42° S together and separate from all other populations to the east in Argentina and to the south regardless of whether they drain into the Atlantic or Pacific Oceans (figure 2). Within this group of north-central Chilean collections, collections line up in a pattern that correlates with latitude with the northernmost populations of Maipo and Nilahue (dark blue in figure 2) separated from the remaining populations further to the south. All populations in this cluster, including Maipo and Nilahue, have their headwaters west of

the Andes with the exception of Neltume and Panguipulli, which are part of the Valdivia river system, the headwaters of which lie east of the Andes in Lake Lácar in Argentina. At the opposite end of the neighbour-joining tree are all populations with headwaters east of the Andes in Argentina. Some of these populations drain into the Atlantic (e.g. collections from the Colorado, Negro, Chubut, Chalia and Santa Cruz rivers) and some into the Pacific Ocean (e.g. Puelo, Futalaufquen/Yelcho, Baker and Pascua river (Lake San Martín)). Two major clusters are identified in this large group. One cluster comprises all collections from northern Patagonia (from north to south: Colorado and Negro rivers, Puelo (lake and river), and lakes Futalaufquen and Musters), while the second cluster comprises all collections from southern Patagonia (e.g. Baker, Pascua and Santa Cruz river systems).

### (c) Analysis of molecular variance

In the first analysis, when populations were grouped according to present-day drainage (Pacific versus Atlantic) irrespective of whether their headwaters lie east or west of the Andes, the percentage of the total variation explained by the variance among groups is 5% (electronic supplementary material, table S3a). By contrast, this percentage was 26.94% when the collections from the trans-Andean systems: Puelo, Futalaufquen/Yelcho, Baker and Pascua were grouped according to their presumed ancestral drainage direction (electronic supplementary material, table S3b) (24.91% if the collections from lakes Neltume and Panguipulli in the Valdivia river system are pooled with the Atlantic drainage group, see below).

We then examined the relative influence of each one of these systems (Puelo, Futalaufquen/Yelcho, Baker and Pascua) on the percentage of variation explained by the variance among groups by grouping them one at a time with the Pacific draining systems. A small decline in the percentage of the variance explained by differences among groups would indicate the system had little influence in the distribution of genetic variance between the Pacific and Atlantic draining groups. In all cases, the percentage of the variance explained by differences among groups dropped considerably from 27% when their ancestral drainage was assumed to have been the Atlantic (electronic supplementary material, table S3b), to 15.45%, 21.75% and 11.23% when grouping each of Puelo, Futalaufquen/Yelcho and Baker individually with all other Pacific draining systems (electronic supplementary material, table S3c–e). The percentage of genetic variation explained by the differences between groups also dropped when pooling the collections from the Pascua River with their contemporary Pacific drainage, but only to 25% (electronic supplementary material, table S3f). This is not surprising given that only 10 individuals were available from a single location (Lake San Martín) within this system.

### (d) Population structure

Structure analyses were conducted at four hierarchical levels. At the highest level, all 16 systems organized by drainage and latitude were analysed together. At this level, $K = 2$ (figure 3a), with the first cluster comprising *P. trucha* individuals from all systems in Chile north of latitude 42° S. These include the collections from Maipo and Nilahue at the northern edge of the species distribution to the collections from the Maullín system in the south. The second cluster comprises all collections from systems with headwaters east of the Andes

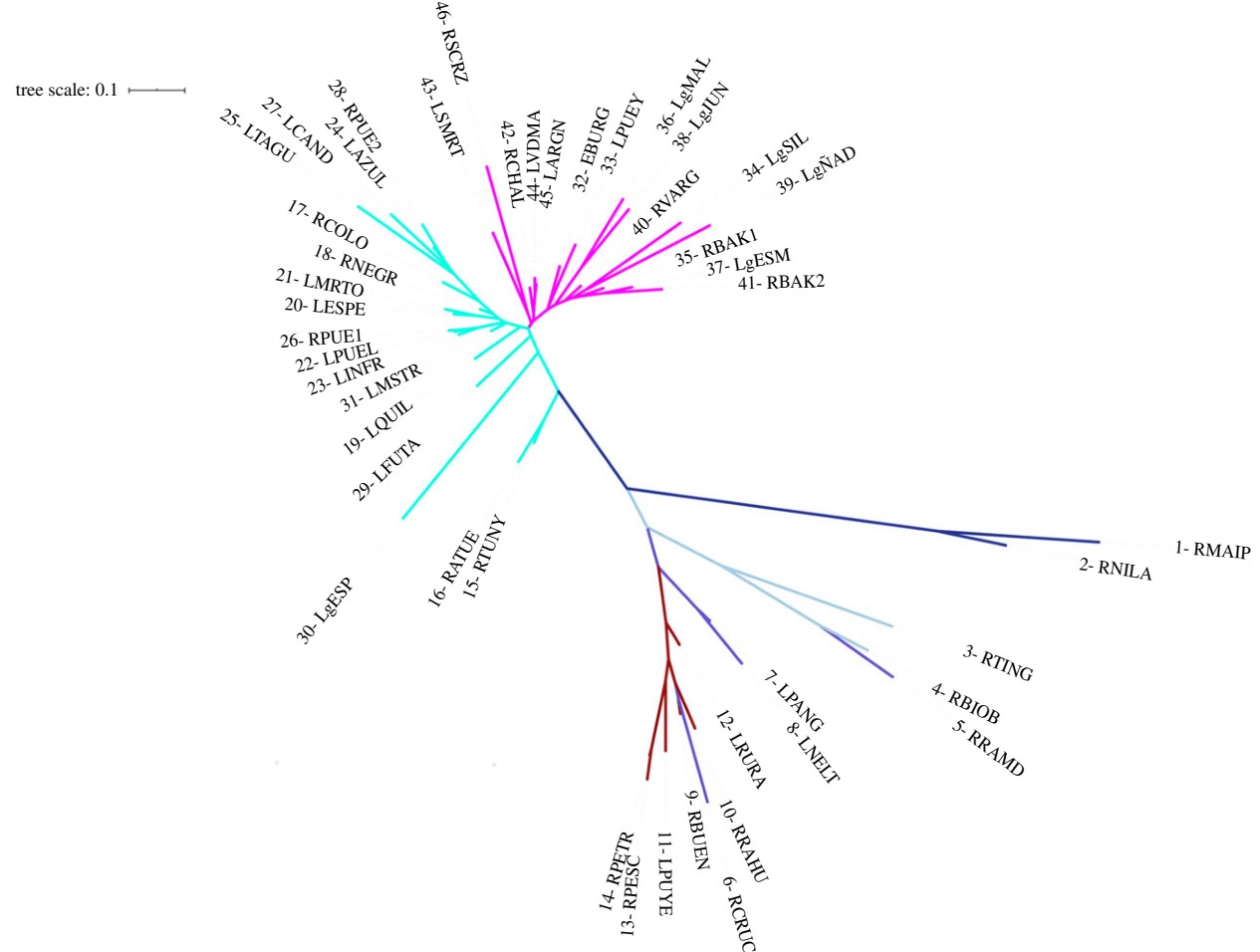

**Figure 2.** Neighbour-joining tree representing *P. trucha* from 16 river systems (46 sampling locations: lakes or rivers) throughout the species distribution in Chile and Argentina. At the broadest scale, the tree distinguishes collections from Pacific draining systems north of 42° S from collections with headwaters east of the Andes that drain into the Atlantic and those that are trans-Andean south of 42° S and drain into the Pacific. Within the first group, collections from Maipo and Nilahue (1 and 2) and to a lesser extent, from the Biobío system (3–5) are distinguishable from those to the south (6–14). Within the second group the colouring reflects two glacial refugia east of Andes, one in northern and one in southern Patagonia. Seven collections have sample sizes $n \leq 2$ (electronic supplementary material, table S1), regardless, they cluster within their respective river systems. (Online version in colour.)

from latitude 33° S to 50° S regardless of contemporary drainage (i.e. whether Pacific or Atlantic; these include: Colorado, Negro, Puelo, Futalaufquen/Yelcho, Chubut, Baker, San Martin and Santa Cruz). Notably, the collections from the northern edge of the species distribution in Chile (Maipo, Nilahue) and in Argentina (Tunuyán, Atuel) appear to exhibit some level of admixture (figure 3a).

The second hierarchical level (figure 3b) examines separately collections from the two clusters above. At this level, *P. trucha* from Maipo and Nilahue (populations 1 and 2) appear genetically distinguishable from those from the Rapel and Biobío systems (populations 3–5) and both groups are distinct from collections form further south including those from the Valdivia, Bueno and Maullín systems (populations 6–14), which at this hierarchical level appear indistinguishable from each other (figure 3b). The populations with headwaters in Argentina (populations 16–46) clustered into two groups, again indicating a latitudinal divide with the more northern river systems (Desaguadero, Colorado, Limay and Negro, Puelo, Futalaufquen, and Chubut) grouping together, and a second grouping composed of the more southern systems (Baker, Chalia, Pascua, and Santa Cruz) (figure 3b). At the third hierarchical level (figure 3c), we found: (i) no discernable difference between fish from the Maipo and Nilahue rivers

with Nilahue containing some migrant and admixed individuals (figure 3c); (ii) significant differences between collections from the Rapel and Biobío systems (populations 3 versus 4 and 5); and (iii) the remaining populations in this pool (populations 6–14) clustering into two groups and thus requiring a fourth hierarchical level of analysis where differences are uncovered among the Valdivia (populations 6–8), Bueno (populations 9–12) and Maullín systems (figure 3d). We also found differences between and among Atlantic and trans-Andean draining systems to the south as well as among some of the sampling locations (lakes or rivers) within each of these systems (figure 3c). Analyses at this fourth hierarchical level were also conducted with collections from the Baker (populations 32–41) and those from the Chalia, Pascua and Santa Cruz river systems (populations 42–46), where genetic differences are manifested among populations within rivers systems probably largely as a result of genetic drift (figure 3d).

## 4. Discussion

In this study, we describe a pattern of genetic divergence among *P. trucha* populations from throughout the species' range reflecting both the role of the Andes as a geographical barrier to

**6**

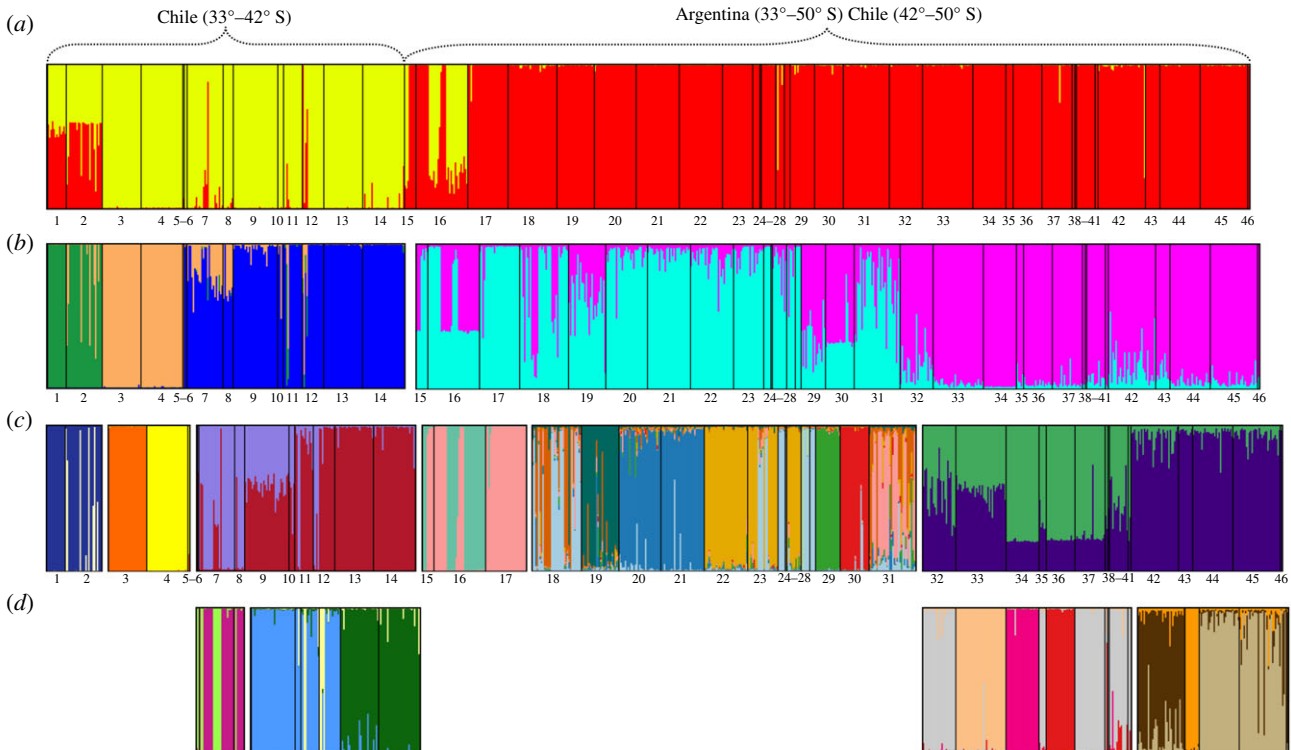

**Figure 3.** Hierarchical population structure analyses of *P. trucha* from 16 drainages. Populations are ordered as in the electronic supplementary material, table S1. (*a*) All sampling locations, $K = 2$ distinguishing collections from Pacific draining systems north of latitude 42°S versus Atlantic draining systems throughout the species' distribution with those from trans-Andean systems south of 42° S. (*b*) Genetic differentiation within each of the groupings found in level (*a*) above. Two groups are distinguished among the collections from the Atlantic draining systems in Argentina and the trans-Andean systems south of 42° S probably reflecting two glacial refugia east of the Andes, in northern southern Patagonia, respectively. Levels (*c*) and (*d*), differences among sampling locations within river systems reflecting influence of genetic drift. Within (*c*) the Baker system (samples 32–41) is distinguishable from Chalia, Pascua and Santa Cruz systems (samples 42–46). Differences among these sampling locations become apparent in (*d*). Lake San Martin (43), drains into both the Atlantic and Pacific. (Online version in colour.)

dispersal and that of the glacial cycles of the Quaternary (2.5 Ma–10 000 BP; [35]) in forcing populations to retreat to glacial refugia and in influencing patterns of recolonization. Our analyses, based on a suite of 53 sequenced nuclear microsatellite markers, indicate that *P. trucha* survived the LGM in at least three glacial refugia. A refugium (or possibly multiple refugia) located west of the Andes and north of where the ice sheets reached the Pacific coast during the LGM (latitude 42° S), served as the source(s) for recolonization of some of the northern river systems west of the Andes. *Percichthys trucha* also survived the LGM in at least two refugia east of the Andes and east and/or north of the ice sheets, one in north-central Patagonia, and the other in southern Patagonia (figure 3*a*,*b*). Atlantic-draining river systems were recolonized from these refugia, and there is a clear distinction between northern systems which were recolonized from the north and/or east, and southern systems which were recolonized from a distinct and probably more southerly refugium. The refugia east of the Andes were also the source of post-glacial colonists for at least five Pacific-draining river systems. These systems all have headwater lakes east of the Andean divide and were probably recolonized during the early stages of deglaciation, when they drained towards the Atlantic. As the glaciers receded, these drainages reversed direction, and currently flow towards the Pacific, but retain *P. trucha* populations derived from the eastern refugia. In fact, these are the only populations of *P. trucha* west of the Andes and south of 42° S, indicating (i) the absence of post-LGM north to south migration into this region, and (ii) that unlike *Galaxias platei* [22] and other more cold-tolerant

species [32,34], *P. trucha* did not persist in small glacial refugia west of the Andes where the ice sheet reached the sea.

The deep differentiation between *P. trucha* populations occupying Atlantic drainages, and those in most Pacific drainages, supports the rise of the Andes (beginning an estimated 23 Ma [28]), as the primary driver of phylogeographic and biogeographic patterns in Patagonian flora and fauna. Since then, phylogeographic patterns have been modified by climatic and geomorphological events, chief of which were the Quaternary glacial cycles [60–61], which caused species ranges to shift, expand and contract, with impacts on genetic diversity as well as distribution and also rearranged Patagonian landscapes and riverscapes ([16,22], this study). The most recent glaciation [60–61] culminated in a 1800 km long ice sheet that covered the Andes from latitude 38° S to 55° S approximately 20 000 BP (figure 1). This ice sheet extended east onto the Patagonian steppe, and west to the Pacific Ocean south of latitude 42° S (figure 1 and [34]). Some species probably survived in local refugia within the glaciated region, while others will have moved north, west or east of the ice sheets. Refugia are known to have existed north and west of the continental ice on the western side of the Andes ([35,62]; figure 1), as well as east and north of the ice sheets on the eastern side ([16] this study). In addition, small refugia appear to have existed within the ice sheet (west of the Andes and south of 42° S) for some species [22,32,33].

Our results demonstrate the power of a relatively large suite of nuclear microsatellite DNA markers to unravel phylogeographic patterns in a widespread freshwater fish, an approach

made possible by the recent development of sequence-based protocols and software for microsatellite genotyping [46]. Such protocols provide the advantage that fragments can be sized precisely with many individuals and loci in a single sequencing run, thus reducing genotyping cost and time and minimizing the need for standardization across laboratories [63–65]. Although microsatellites are sequenced, in the present study alleles were scored solely based on the length of the repeated motif using the MEGASAT [46] platform. Any potential allelic diversity stemming from the existence of single nucleotide polymorphisms (SNPs) within the repeated motifs has thus remained undetected. In the future, the use of sequence-based simple sequence repeats (SSR) that incorporate SNP information within the repeat motif are likely to improve population genetics and phylogenetic analyses and inferences. Regardless, using a combination of a large panel of genetic markers and landscape information reflecting glacial history, we describe phylogeographic patterns for one of the most widespread species of fish in Patagonia, *P. trucha*. These patterns are consistent with the presence at least two glacial refugia east of the Andes during the LGM, and at least one, and probably more refugia west of the Andes in central Chile (north of 42° S where the ice did not reach the sea). Our study also provides genetic evidence that is consistent with geographically separated and multiple episodes of drainage reversal that influenced the distribution of genetic diversity of *P. trucha* throughout most of its latitudinal range. We discuss the evidence for an influence of drainage reversal on patterns of genetic diversity and species distribution in more detail below.

## (a) Headwaters west of the highest Andean peaks

Populations of *P. trucha* currently inhabiting river systems with their headwaters west of the Andean divide are all found north of 42° S, where the ice sheet did not reach the Pacific coast during the LGM [35]. These populations either persisted in local refugia (possibly downstream within the same river system) or these systems must have been recolonized from the north. The four northernmost populations, those from the rivers Maipo and Nilahue (samples 1 and 2 in the electronic supplementary material, table S1), and those from the Rapel and Valdivia systems (samples 3–5) were genetically distinguishable from the other Pacific draining systems (figures 2 and 3), suggesting that these four populations probably survived in refugia separate from the rest. The relative geographical isolation of these river systems, particularly that of the Maipo and Nilahue, has probably helped the populations maintain their distinctiveness (figure 1).

Among the other Pacific draining systems with headwaters west of the Andes, we find less genetic differentiation than for the northern four systems but more than is seen among the Atlantic draining systems (note branch lengths in figure 2). In addition, there is a shallow signal of differentiation that follows a latitudinal gradient from north to south. We infer, therefore, that the populations in these Pacific draining systems probably persisted through the LGM in deglaciated coastal areas, but that they were also influenced by predominantly north to south gene flow. North to south recolonization on the western side of the Andes has been observed in terrestrial species, such as the forest-dwelling mouse *Abrothrix olivaceus* and the steppe-dwelling mouse *Abrothrix xanthorhinus* [66]. Gene flow among aquatic populations occupying different watersheds may have occurred during periods of high discharge as the ice sheet melted.

## (b) Headwaters east of the highest Andean peaks

River systems with headwaters east of the Andes that harbour *P. trucha* comprise all major river systems that currently drain into the Atlantic Ocean (the Colorado, Negro, Chubut and Santa Cruz river systems) and several Pacific-draining, trans-Andean systems (the Puelo, Futalaufquen/Yelcho, Baker and Pascua river systems) (figure 1; electronic supplementary material, table S1). These populations formed two genetically differentiated groups, suggesting that they were colonized from two separate glacial refugia. The *P. trucha* populations currently found in the Desaguadero, Colorado, Negro and Puelo river systems form a group, and probably originated from a northwestern refugium, while populations in the Chubut, Santa Cruz, Futalaufquen/Yelcho, Baker, Pascua and Chalia catchments probably originate from a southern refugium. The presence of southern refugia has also been proposed for some terrestrial taxa [12,31,67]. Southern refugia were probably facilitated by the greatly expanded Patagonian steppe area east of the ice sheet, with much of the continental shelf exposed as sea-level fell during glacial periods [68,69], and the persistence of such refugia may have been enhanced by the high connectivity among drainages afforded by the braided and deltaic connections that formed and re-formed on the shelf during glacial periods [69,70].

Thus, the northern and southern genetic groupings both contain populations of *P. trucha* from present-day Atlantic drainages and populations in trans-Andean systems, where the headwaters lie east of the Andes, but the system drains west to the Pacific (figure 3b,c). There is strong evidence from geomorphological studies of the Baker [38,40–41] trans-Andean system for drainage reversal following the LGM. At the LGM, the lakes at the headwaters of the Baker system were part of a large eastward draining proglacial lake (Lake Chalenko) that extended from approximately latitude 46° S to 48° S, with raised deltas tens to hundred of metres above contemporary valley floors still visible [36,37,39,71]. As the glaciers melted, the ice dam that had formed the western limit of the lake was breached (*ca* 12 000 BP [38]), and the water began flowing west, through valleys previously blocked by ice ([35,38–40] and references therein). The genetic similarity of *P. trucha* populations in the Baker system with populations in Atlantic-draining systems at similar latitudes is thus consistent with the geomorphological findings. In this study, we have identified three additional river systems that have probably undergone post-LGM drainage reversals: the Puelo, Futalaufquen/Yelcho, and the Pascua systems. All are trans-Andean systems, Pacific drainages with headwaters east of the Andes, and all contain populations of *P. trucha* that resemble populations from Atlantic drainages at similar latitudes. It is known that proglacial lakes extended over broad latitudinal ranges (42–49° S and 51–53° S) [38,40,41,71–74], but to our knowledge, detailed geomorphological studies of the type conducted for the Baker system have not been done for the other three.

*Percichthys trucha* inhabiting the trans-Andean Valdivia river system (latitude approx. 40° S, headwaters in Lake Lácar in Argentina) comprise an exception to the pattern of higher genetic similarity with Atlantic than with Pacific draining systems described above for the other four southern trans-Andean systems (Puelo, Futalaufquen/Yelcho, Baker, and Pascua). Grouping the collections from the Valdivia system (i.e. Panguipulli and Neltume) with those of Atlantic drainage decreased the percentage of genetic variation explained by

groups (from approx. 27% to 25%) suggesting that *P. trucha* inhabiting these lakes originate from ancestral populations that survived in local refugia, probably preventing the dispersal and expansion of *P. trucha* from east of the Andean highest peaks by a 'Founder takes all' effect [75].

South of latitude 42° S, where the ice sheet reached the Pacific during the LGM, there are no *P. trucha* populations outside of trans-Andean river systems. Drainage reversals are thus the only mechanism by which *P. trucha* was able to colonize this region after the glaciers receded, and there appears to have been no gene flow between drainages since that time. Drainage reversals are thus responsible, not only for patterns of genetic diversity, but also for the current geographical range of the species. This pattern of distribution is shared with at least one other native Patagonian fish, a catfish of the family *Diplomystidae* [30], providing further evidence of the importance for the biogeographic patterns of aquatic taxa in Patagonia of the late Pleistocene glacial melt induced drainage reversals.

In summary, we have presented evidence based on a suite of 53 sequenced nuclear microsatellite DNA markers that, after the LGM, *P. trucha*, a widespread Patagonian fish, colonized the western side of the Andes from the eastern side, and that this occurred in at least four distinct river systems, the Puelo, the Futalaufquen/Yelcho, the Baker, and the Pascua systems. While detailed geomorphological evidence exists only for the Baker system [38], drainage reversals during the retreat of the glaciers appear to be the most likely mechanism for all. In addition, we describe evidence for at least three glacial refugia for *P. trucha*, one (possibly more) west of the Andes north of 42° S, and the other two, east of the Andes, the first in northwestern and northern Patagonia and the second on the southern Patagonian steppe. Conversely, we find no evidence that *P. trucha* survived the LGM west of the Andes south of 42° S. The presence of *P. trucha* populations west of the Andes in these southern regions is a consequence of dispersal from the east, most likely owing to drainage reversals that took place in several river systems following the LGM. Our study, therefore, highlights the synergistic value of combining genetic data from a large panel of sequenced microsatellite DNA with information on landscape evolution to develop and test biogeographic hypotheses.

**Ethics.** Field collections in both Argentina and Chile were conducted under national, provincial and National Parks permits as appropriate.

**Data accessibility.** Microsatellite genotypes are available in the Dryad Digital Repository (https://doi.org/10.5061/dryad.n8pk0p2s5) [76].

**Authors' contributions.** D.E.R., E.H. and S.J.W. participated in the field-work. G.R.M. developed and tested the 75 microsatellite markers. A.P.S. and G.R.M. conducted the molecular laboratory work. D.E.R., A.P.S. and G.R.M. performed analysis. D.E.R. wrote the paper with input from all authors.

**Competing interests.** Authors declare no competing interests.

**Funding.** We thank the Committee for Research and Exploration of the National Geographic Society, Washington, for generous support for fieldwork in 2001 (NGS 6799-00) and 2007 (NGS 8168-07), and NSERC Discovery grants and a Special Research Opportunities award (SROPJ/326493-06), as well as Universidad de Concepción (DIUC-Patagonia 205.310.042-ISP) and FONDECYT (no. 1080082) grants which are gratefully acknowledged. Some samples were collected with support from a 2006 to 2010 NSF-PIRE award (OISE 0530267).

**Acknowledgements.** We thank the numerous colleagues from Universidad de Concepción, Universidad Nacional del Comahue (Argentina) and Dalhousie University who assisted with sample collections during more than two decades of fieldwork in Patagonia.

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
