## [Reviewer comments · Proceedings of the Royal Society B: Biological Sciences]

Review History

RSPB-2020-0468.R0 (Original submission)

Review form: Reviewer 1

Recommendation

Major revision is needed (please make suggestions in comments)

Scientific importance: Is the manuscript an original and important contribution to its field?

Good

General interest: Is the paper of sufficient general interest?

Good

Quality of the paper: Is the overall quality of the paper suitable?

Good

Is the length of the paper justified?

Yes

Should the paper be seen by a specialist statistical reviewer?

No

Do you have any concerns about statistical analyses in this paper? If so, please specify them explicitly in your report.

Yes

It is a condition of publication that authors make their supporting data, code and materials available - either as supplementary material or hosted in an external repository. Please rate, if applicable, the supporting data on the following criteria.

Is it accessible?

Yes

Is it clear?

Yes

Is it adequate?

Yes

Do you have any ethical concerns with this paper?

No

Comments to the Author

Please, find below my review of the manuscript RSPB-2020-0468 entitled “Across the Andes on fin II: Multiple drainage reversal episodes and glacial refugia in a Patagonian fish revealed by sequenced microsatellites” by Dr. Daniel Ruzzante and collaborators.

By analyzing a sequenced microsatellite genotypic database (53 loci) concerning different *Percichthys trucha* populations sampled in Chile and Argentina (46 locations, 16 river systems), the authors examine the phylogeography of this species and the role of drainage reversals during deglaciations following the last glacial maximum on shaping current patterns of genetic diversity of this species in Patagonia. For that, they coupled three methods: a hierarchical clustering approach, the built of a neighbor-joining tree based on Nei’s *D_a* genetic distances, and hierarchical analyses of molecular variances (AMOVAs).

This study is interesting, original, well-written and the analytical & statistical treatment is appropriate & robust. It nicely illustrates how phylogeographic and geomorphological may be combined to explore past processes having shaped species distributions and their spatial patterns of genetic diversity. I have however some concerns that must be addressed by the authors to have their MS reach the quality standard necessary for being published in PROCS B.

I hope that these suggestions may help improving the robustness and the reach of the MS.

(1) The authors use sequencing-based SSRs to conduct phylogeographic analyses (notably the NJ tree analysis) and to link genetic patterns with geomorphological/biogeographical processes like drainage reversals. I’m OK with that, especially given the relatively-recent timeframe covered by the authors’ interpretation (from Last Glacial Maximum to current dates). However, many readers may wonder why the authors didn’t use mitochondrial loci to perform this study and may be less convinced about the validity/robustness of authors’ SSR-based interpretations. It is thus necessary that the authors provide some discussion justifying the validity/usefulness of using sequencing-based SSRs for phylogeography instead of (i) using “traditional” mitochondrial loci used in phylogeography and (ii) using classical electrophoresis-based SSR loci.

(2) The study species (*Percichthys trucha*) is unknown for many people. Please, provide one or two sentences presenting this species in the Materials & methods section. As diversity at microsatellite loci is particularly sensitive to human impacts, it will be important to state whether this species is harvested, supplemented with hatchery-reared, or if it is the target of recreational/self-consuming angling by local populations. This may allow readers to assess at which point current patterns are solely due to natural/biogeographical/geomorphological

processes and drivers.

(3) L.76-77: “a suite of 53 sequenced nuclear microsatellite DNA markers”: are there analytical advantages of using sequenced-based SSRs compared to classical ones in your study (e.g. obtaining a high number of genotyped loci, reducing homoplasy etc...)? If yes, this must be clearly stated, as it will be one of the strengths of the papers.

(4) I'm not fully convinced by the title of the article “Across the Andes on fin II: Multiple drainage reversal episodes and glacial refugia in a Patagonian fish revealed by sequenced microsatellites”. Being a two-part title, I found it too much longer. I recommend keeping only the second part to be more direct, clear and punchy. Further, the first part of the title may generate confusion. For instance, when I started reading the title, I directly thought that this MS was directly related to another recently-published paper on the same species & expected strong interconnection between both papers. It appears that “Across the Andes on Fin I” paper was published 12 years ago, and concerned another species. Finally, the title clearly states the use of “sequenced microsatellites”. Again, I think that the advantages of using this type of loci must be discussed in the paper, especially if their use is specifically mentioned in the title.

(5) Concerning loci filtering procedures: given the high number of loci considered in the study, it is probable that some of them may be linked or being affected by selection, hence potentially affecting subsequent statistical analyses. Linkage disequilibrium per population pairs per loci should be assessed, and loci should be removed accordingly if they display recurrent signals of significant linkage across populations. Further, loci should be tested for neutrality, and all loci displaying evidence of selection must be removed from main analyses.

(6) One advantage of sequence-based SSRs vs. classical electrophoresis-based SSRs is the reduction of homoplasy i.e. two alleles may be indexed as identical (i.e. same length of the amplified marker) when using traditional electrophoresis-based SSR genotyping, but may be actually different alleles if one or more SNPs occur in the sequence. By obtaining full SSR sequences for each individual, these SNP may be detected, hence revealing allelic diversity that would have remained hidden if classical SSR genotyping had been used. This gain of information and diversity is particularly interesting for phylogeography and genetic inference procedures. Has this potential source of variation been accounted for in this study or in the MEGASAT procedure? or have the alleles being scored solely based on the length of the repeated motif instead? If the second option is the true one, I recommend adding some sentences in the discussion drawing some perspectives of how using sequence-based SSR may improve phylogenetic and population-genetics analyses in the future, as we will probably see a new wave of microsatellite-based studies in the near future due to the updating of SSR-loci development using modern sequencing procedures (e.g. Lepais et al 2019 biorxiv doi: 10.1101/649772).

(7) The principle & procedure of hierarchical STRUCTURE analyses must be clearly explained in the Materials & Methods section. Also, the criterion used to choose the best K at each step must be stated (is the delta K of Evanno or other criteria has been used?).

(8) At least 14 out of 46 sampling sites have less than 10 individuals, with some having 1 individual per location. Given that the NJ tree building requires calculating genetic distances (Da) among populations/sampling sites, is it reliable to include populations with low sample sizes in this analysis? Consider removing them or justify why you keep them in the analyses.

Review form: Reviewer 2

Recommendation

Accept as is

Scientific importance: Is the manuscript an original and important contribution to its field?
Excellent

General interest: Is the paper of sufficient general interest?
Excellent

Quality of the paper: Is the overall quality of the paper suitable?
Excellent

Is the length of the paper justified?
Yes

Should the paper be seen by a specialist statistical reviewer?
No

Do you have any concerns about statistical analyses in this paper? If so, please specify them explicitly in your report.
No

It is a condition of publication that authors make their supporting data, code and materials available - either as supplementary material or hosted in an external repository. Please rate, if applicable, the supporting data on the following criteria.

Is it accessible?
Yes

Is it clear?
Yes

Is it adequate?
Yes

Do you have any ethical concerns with this paper?
No

Comments to the Author

An excellent contribution. The manuscript brings relevant information to understand the transandean distribution of freshwater organisms in the southern portion of the continent. It is relevant and can be tested with other organisms besides fish.

Decision letter (RSPB-2020-0468.R0)

14-Apr-2020

Dear Dr Ruzzante:

Your manuscript has now been peer reviewed and the reviews have been assessed by an Associate Editor. The reviewers' comments (not including confidential comments to the Editor) and the comments from the Associate Editor are included at the end of this email for your reference. As you will see, the reviewers and the Editors have raised some concerns with your manuscript and we would like to invite you to revise your manuscript to address them.

Research ethics:

Use of animals and field studies:

Please submit a copy of your revised paper within three weeks. If we do not hear from you within this time your manuscript will be rejected. If you are unable to meet this deadline please let us know as soon as possible, as we may be able to grant a short extension.

Best wishes,
Dr Daniel Costa
mailto:proceedingsb@royalsociety.org

Associate Editor

Board Member: 1

Comments to Author:

As can be gauged in both of these reviews, the Referees found that this study investigating how past processes have shaped species distributions and their spatial patterns of genetic diversity, could be of potentially high interest to PRSB. In particular, the novelty of the microsatellite sequencing approach combined with phylogeographic geomorphological methods in an iconic system was interesting and original. Nonetheless, in my own reading I agree with the Referees that there is some additional work required to really take advantage of the data and system while addressing some underlying concerns of how these data may be influencing the conclusions of the study. First, I agree that the merits of the microsatellite data under this experimental design need to be more explicit, both in respect of their justification compared to more traditional phylogeographical methods (e.g., mtDNA) and taking full advantage of the analytical “opportunities” to understand the underlying genetic diversity of these microsatellites (homoplasy, gametic phase disequilibrium, selection) and how these may be impacting the data and results. Along these lines, the Referees raised some concerns about the impact of small sample sizes for certain sites. While sample size was unlikely to influence the main findings of the paper, some justification is merited here. Otherwise, the Referees make several other useful suggestions to clarify and broaden the scope of the paper and these should each be addressed in the revision. Overall, while I am confident that these revisions will improve the robustness of the MS, they do reflect significant revisions that may have an impact on the corresponding results and conclusions of the study.

Reviewer(s)' Comments to Author:

Referee: 1

Comments to the Author(s)

Please, find below my review of the manuscript RSPB-2020-0468 entitled “Across the Andes on fin II: Multiple drainage reversal episodes and glacial refugia in a Patagonian fish revealed by sequenced microsatellites” by Dr. Daniel Ruzzante and collaborators.

By analyzing a sequenced microsatellite genotypic database (53 loci) concerning different *Percichthys trucha* populations sampled in Chile and Argentina (46 locations, 16 river systems), the authors examine the phylogeography of this species and the role of drainage reversals during deglaciations following the last glacial maximum on shaping current patterns of genetic diversity of this species in Patagonia. For that, they coupled three methods: a hierarchical clustering approach, the built of a neighbor-joining tree based on Nei's D_a genetic distances, and hierarchical analyses of molecular variances (AMOVAs).

This study is interesting, original, well-written and the analytical & statistical treatment is appropriate & robust. It nicely illustrates how phylogeographic and geomorphological may be combined to explore past processes having shaped species distributions and their spatial patterns of genetic diversity. I have however some concerns that must be addressed by the authors to have their MS reach the quality standard necessary for being published in PROCS B. I hope that these suggestions may help improving the robustness and the reach of the MS.

(1) The authors use sequencing-based SSRs to conduct phylogeographic analyses (notably the NJ tree analysis) and to link genetic patterns with geomorphological/biogeographical processes like drainage reversals. I'm OK with that, especially given the relatively-recent timeframe covered by the authors' interpretation (from Last Glacial Maximum to current dates). However, many readers may wonder why the authors didn't use mitochondrial loci to perform this study and may be less convinced about the validity/robustness of authors' SSR-based interpretations. It is thus necessary that the authors provide some discussion justifying the validity/usefulness of using sequencing-based SSRs for phylogeography instead of (i) using "traditional" mitochondrial loci used in phylogeography and (ii) using classical electrophoresis-based SSR loci.

(2) The study species (*Percichthys trucha*) is unknown for many people. Please, provide one or two sentences presenting this species in the Materials & methods section. As diversity at microsatellite loci is particularly sensitive to human impacts, it will be important to state whether this species is harvested, supplemented with hatchery-reared, or if it is the target of recreational/self-consuming angling by local populations. This may allow readers to assess at which point current patterns are solely due to natural/biogeographical/geomorphological processes and drivers.

(3) L.76-77: "a suite of 53 sequenced nuclear microsatellite DNA markers": are there analytical advantages of using sequenced-based SSRs compared to classical ones in your study (e.g. obtaining a high number of genotyped loci, reducing homoplasy etc...)? If yes, this must be clearly stated, as it will be one of the strengths of the papers.

(4) I'm not fully convinced by the title of the article "Across the Andes on fin II: Multiple drainage reversal episodes and glacial refugia in a Patagonian fish revealed by sequenced microsatellites". Being a two-part title, I found it too much longer. I recommend keeping only the second part to be more direct, clear and punchy. Further, the first part of the title may generate confusion. For instance, when I started reading the title, I directly thought that this MS was directly related to another recently-published paper on the same species & expected strong interconnection between both papers. It appears that "Across the Andes on Fin I" paper was published 12 years ago, and concerned another species. Finally, the title clearly states the use of "sequenced microsatellites". Again, I think that the advantages of using this type of loci must be discussed in the paper, especially if their use is specifically mentioned in the title.

(5) Concerning loci filtering procedures: given the high number of loci considered in the study, it is probable that some of them may be linked or being affected by selection, hence potentially affecting subsequent statistical analyses. Linkage disequilibrium per population pairs per loci should be assessed, and loci should be removed accordingly if they display recurrent signals of significant linkage across populations. Further, loci should be tested for neutrality, and all loci displaying evidence of selection must be removed from main analyses.

(6) One advantage of sequence-based SSRs vs. classical electrophoresis-based SSRs is the reduction of homoplasy i.e. two alleles may be indexed as identical (i.e. same length of the amplified marker) when using traditional electrophoresis-based SSR genotyping, but may be actually different alleles if one or more SNPs occur in the sequence. By obtaining full SSR sequences for each individual, these SNP may be detected, hence revealing allelic diversity that would have remained hidden if classical SSR genotyping had been used. This gain of information and diversity is particularly interesting for phylogeography and genetic inference procedures. Has this potential source of variation been accounted for in this study or in the MEGASAT procedure? or have the alleles being scored solely based on the length of the repeated motif instead? If the second option is the true one, I recommend adding some sentences in the discussion drawing some perspectives of how using sequence-based SSR may improve phylogenetic and population-genetics analyses in the future, as we will probably see a new wave of microsatellite-based studies in the near future due to the updating of SSR-loci development using modern sequencing procedures (e.g. Lepais et al 2019 biorxiv doi: 10.1101/649772).

(7) The principle & procedure of hierarchical STRUCTURE analyses must be clearly explained in the Materials & Methods section. Also, the criterion used to choose the best K at each step must be stated (is the delta K of Evanno or other criteria has been used?).

(8) At least 14 out of 46 sampling sites have less than 10 individuals, with some having 1 individual per location. Given that the NJ tree building requires calculating genetic distances (Da) among populations/sampling sites, is it reliable to include populations with low sample sizes in this analysis? Consider removing them or justify why you keep them in the analyses.

Referee: 2

Comments to the Author(s)

An excellent contribution. The manuscript brings relevant information to understand the transandean distribution of freshwater organisms in the southern portion of the continent. It is relevant and can be tested with other organisms besides fish.

Author's Response to Decision Letter for (RSPB-2020-0468.R0)

See Appendix A.

Decision letter (RSPB-2020-0468.R1)

05-May-2020

Dear Dr Ruzzante

I am pleased to inform you that your manuscript entitled "Multiple drainage reversal episodes and glacial refugia in a Patagonian fish revealed by sequenced microsatellites" has been accepted for publication in Proceedings B.

You can expect to receive a proof of your article from our Production office in due course, please check your spam filter if you do not receive it. PLEASE NOTE: you will be given the exact page

length of your paper which may be different from the estimation from Editorial and you may be asked to reduce your paper if it goes over the 10 page limit.

Open Access

Paper charges

Sincerely,

Dr Daniel Costa

Associate Editor:

Board Member

Comments to Author:

(There are no comments.)

Appendix A

Dear Dr Daniel Costa, Editor

Enclosed please find our revised MS now entitled: “Multiple drainage reversal episodes and glacial refugia in a Patagonian fish revealed by sequenced microsatellites”. We thank both referees. Referee 1 provided very insightful comments and we particularly thank her/him for them. We have found these comments objective and insightful. They helped raise the quality and we suspect, potential reach of our paper. We have addressed each and every one of them. Below are the 8 points raised by referee 1 and our responses to them (in yellow highlight)

1) The authors use sequencing-based SSRs to conduct phylogeographic analyses (notably the NJ tree analysis) and to link genetic patterns with geomorphological/biogeographical processes like drainage reversals. I’m OK with that, especially given the relatively-recent timeframe covered by the authors’ interpretation (from Last Glacial Maximum to current dates). However, many readers may wonder why the authors didn’t use mitochondrial loci to perform this study and may be less convinced about the validity/robustness of authors’ SSR-based interpretations. It is thus necessary that the authors provide some discussion justifying the validity/usefulness of using sequencing-based SSRs for phylogeography instead of (i) using “traditional” mitochondrial loci used in phylogeography and (ii) using classical electrophoresis-based SSR loci.

ANSWER: We addressed these comments in several places in the MS (See also answers to points 3 and 6). In the Introduction we wrote:

Lines 77-84:

“The use of a high number of sequenced nuclear microsatellite markers provides two advantages: (1) it addresses the general concern common to most phylogeographic studies that patterns based on mtDNA polymorphism reflect the coalescence time and evolutionary history of a single gene and not necessarily that of the organism a whole, and (2) their relatively high mutation rate resulting in high polymorphism makes microsatellite markers particularly useful for phylogeographic studies that both, focus on relatively short time frames (e.g. Late Pleistocene-Holocene to present) and examine patterns at a finer geographic scale than may often be feasible with mtDNA.”

(2) The study species (*Percichthys trucha*) is unknown for many people. Please, provide one or two sentences presenting this species in the Materials & methods section. As diversity at microsatellite loci is particularly sensitive to human impacts, it will be important to state whether this species is harvested, supplemented with hatchery-reared, or if it is the target of recreational/self-consuming angling by local populations. This may allow readers to assess at which point current patterns are solely due to natural/biogeographical/geomorphological processes and drivers.

ANSWER: We added a paragraph describing the species –

Lines 97-103:

“*Percichthys trucha*, a freshwater fish species native to Patagonia and neighboring regions to the north, is widely distributed throughout the region. The species is not harvested nor is it the focus of recreational angling, which in Patagonia focuses on introduced salmonids. The species has not ever been the subject of supplementation with hatchery reared fish in any of the river systems visited for this study. Current phylogeographic and genetic diversity patterns can thus be safely considered to be solely the result of natural, biogeographical and geomorphological processes and drivers.”

(3) L.76-77: “a suite of 53 sequenced nuclear microsatellite DNA markers”: are there analytical advantages of using sequenced-based SSRs compared to classical ones in your study (e.g. obtaining a high number of genotyped loci, reducing homoplasy etc...)? If yes, this must be clearly stated, as it will be one of the strengths of the papers.

ANSWER: Please see out response to point (1) above and point (6) below.

(4) I’m not fully convinced by the title of the article “Across the Andes on fin II: Multiple drainage reversal episodes and glacial refugia in a Patagonian fish revealed by sequenced microsatellites”. Being a two-part title, I found it too much longer. I recommend keeping only the second part to be more direct, clear and punchy. Further, the first part of the title may generate confusion. For instance, when I started reading the title, I directly thought that this MS was directly related to another recently-published paper on the same species & expected strong interconnection between both papers. It appears that “Across the Andes on Fin I” paper was published 12 years ago, and concerned another species. Finally, the title clearly states the use of “sequenced microsatellites”. Again, I think that the advantages of using this type of loci must be discussed in the paper, especially if their use is specifically mentioned in the title.

ANSWER: Done! The title now reads:

“Multiple drainage reversal episodes and glacial refugia in a Patagonian fish revealed by sequenced microsatellites”

(5) Concerning loci filtering procedures: given the high number of loci considered in the study, it is probable that some of them may be linked or being affected by selection, hence potentially affecting subsequent statistical analyses. Linkage disequilibrium per population pairs per loci should be assessed, and loci should be removed accordingly if they display recurrent signals of significant linkage across populations. Further, loci should be tested for neutrality, and all loci displaying evidence of selection must be removed from main analyses.

ANSWER: We have now conducted tests of LD between pairs of loci in each of 9 collections for which $N \geq 30$. There was no instance of recurrent and significant LD after FDR correction.

We also used BayesAss to test for evidence of selection. To do this and given the high likelihood of false positives with these tests (See Narum and Hess 2011) we tested all 46 collections, the 9 collections with $N \geq 30$, the 4 collections within the Limay-Negro system (all $N \geq 26$) and the two collections within the Maullín system (all $N \geq 27$). While 18, 9, 1 and 0 loci showed up as outliers in each of these tests,

respectively, there was no consistency in the identity of the outlier loci across more than two of these four tests. In fact, the single locus showing up as outlier in the test involving 4 collections within the Limay-Negro system does not show up in the second test. No locus appears as outlier in the comparison between the two samples in the Maullín system. We conclude there is no meaningful evidence of selection in any of these loci.

In the text we addressed both the LD and the selection points as follows:

Lines 173-184:

Genetic quality control

Although the entire data set consisted of $N=835$ individuals genotyped at a panel of 53 microsatellite loci, sample sizes per population were relatively small (mean $N=18.2$, median $N=20.5$). We tested for LD between pairs of loci in the 9 collections available with $N \geq 30$ (see Table 1). We found no evidence of LD that was significant and consistent across the 9 populations tested. We also tested for evidence of selection using all 46 collections, and subsets with relatively large sample sizes. These included the 9 collections with $N \geq 30$, the 4 collections within the Limay-Negro system (all $N \geq 26$) and the two collections within the Maullín system (both $N \geq 27$). While 18, 9, 1 and 0 loci showed up as outliers in each of these tests, respectively, there was no consistency in the identity of the outlier loci across more than two of these four tests. We conclude there is no meaningful evidence of selection in any of these loci and all were retained in all subsequent analyses.

(6) One advantage of sequence-based SSRs vs. classical electrophoresis-based SSRs is the reduction of homoplasy i.e. two alleles may be indexed as identical (i.e. same length of the amplified marker) when using traditional electrophoresis-based SSR genotyping, but may be actually different alleles if one or more SNPs occur in the sequence. By obtaining full SSR sequences for each individual, these SNP may be detected, hence revealing allelic diversity that would have remained hidden if classical SSR genotyping had been used. This gain of information and diversity is particularly interesting for phylogeography and genetic inference procedures. Has this potential source of variation been accounted for in this study or in the MEGASAT procedure? or have the alleles being scored solely based on the length of the repeated motif instead? If the second option is the true one, I recommend adding some sentences in the discussion drawing some perspectives of how using sequence-based SSR may improve phylogenetic and population-genetics analyses in the future, as we will probably see a new wave of microsatellite-based studies in the near future due to the updating of SSR-loci development using modern sequencing procedures (e.g. Lepais et al 2019 biorxiv doi: 10.1101/649772).

ANSWER: We addressed this comment in several places in the MS; in particular, in the Discussion:

(a) Lines 295-305:

“Our results demonstrate the power of a relatively large suite of nuclear microsatellite DNA markers to unravel phylogeographic patterns in a widespread freshwater fish, an approach made possible by the recent development of sequence-based protocols and software for microsatellite

genotyping [49]. Such protocols provide the advantage that fragments can be sized precisely with many individuals and loci in a single sequencing run thus reducing genotyping cost and time and minimizing the need for standardization across laboratories [66-68]. Although microsatellites are sequenced, in the present study alleles were scored solely based on the length of the repeated motif using the MEGASAT [49] platform. Any potential allelic diversity stemming from the existence of SNPs within the repeated motifs has thus remained undetected. In the future, the use of sequence-based simple sequence repeats (SSR) that incorporate SNP information within the repeat motif are likely to improve population genetics and phylogenetic analyses and inferences.”

See also the following lines:

L 260-261:

“Our analyses based on a suite of 53 sequenced nuclear microsatellite markers indicate that *P. trucha* survived the Last Glacial Maximum (LGM) in at least three glacial refugia.”

L 390-391:

“In summary, we have presented evidence based on a suite of 53 sequenced nuclear microsatellite DNA markers that, after the last glacial maximum (LGM), *Percichthys trucha*, a wide-spread Patagonian fish, colonized”

L 403-405:

Our study therefore highlights the synergistic value of combining genetic data from a large panel of sequenced microsatellite DNA with information on landscape evolution to develop and test biogeographic hypotheses.

(7) The principle & procedure of hierarchical STRUCTURE analyses must be clearly explained in the Materials & Methods section. Also, the criterion used to choose the best K at each step must be stated (is the delta K of Evanno or other criteria has been used?).

ANSWER: We addressed this comment by rewriting the appropriate section in M&M as follows.

See lines 139-145:

“We used the program STRUCTURE 2.3.4 [52] to assess population structure hierarchically. We first examined the entire data set. Identified clusters were then independently subject to further STRUCTURE analysis. STRUCTURE was run using 500,000 MCMC permutations and 100,000 burn-in steps with each K value replicated 5 times. We used the Evanno method [53] as implemented in STRUCTURE HARVESTER v0.6.92 [54] to determine the most likely number of clusters. This process was continued up to the identification of individual rivers or lakes. If needed geographic location was also taken into consideration.

(8) At least 14 out of 46 sampling sites have less than 10 individuals, with some having 1 individual per location. Given that the NJ tree building requires calculating genetic distances (D_a) among

populations/sampling sites, is it reliable to include populations with low sample sizes in this analysis? Consider removing them or justify why you keep them in the analyses.

ANSWER: Indeed, some collections have fewer than $N=10$ individuals but the fact that these populations cluster in the expected locations in fig 2 suggest the small sample sizes do not affect the general clustering patterns in the phylogeographic tree. We suspect the high number of microsatellite loci makes up for the small N . To address this point, we focused on the 7 collections with $N \leq 2$. These collections are (See Table S1):

Biobío system

(5) Ramadillas (RRAMD) ($N=1$)

Valdivia system

(6) Rio Cruces (RCRUC) ($N=2$)

Puelo system

(25) Tagua Tagua lake (LTAGU) ($N=1$)

Baker system

(38) Lag Juncal (LgJUN) ($N=2$)

(39) Lag. Los Ñadis (LgÑAD) ($N=1$)

(41) Rio Baker (RBAK2) ($N=2$)

Santa Cruz river system

(46) Sta Cruz river (RSCRUZ) ($N=1$)

A close examination of fig 2 reveals that in virtually all instances these collections cluster within their corresponding systems. There is one potential exception (RCRUC from the Valdivia system) which though in the same branch as the other collections from the Valdivia system, clusters closer to the samples from the neighboring rivers to the south, which are geographically quite close (See inset for samples 6 and 9 in Fig.1).

In the legend to fig 2 we state: Seven collections have sample sizes $N \leq 2$ (Table S1), regardless, they cluster within their respective river systems (See ESM for further details).